

# Mesoscopic electron transport and atomic gases, a review of Frank W. J. Hekking's scientific work

Luigi Amico[1], Denis M. Basko[2], Sebastián Bergeret[3], Olivier Buisson[4], Hervé Courtois[4*],
Rosario Fazio[5,6], Wiebke Guichard[4], Anna Minguzzi[2], Jukka Pekola[7], Gerd Schön[8]

**1** Dipartimento di Fisica e Astronomia, Universita Catania,
Via S. Sofia 64, I-95123 Catania, Italy.
**2** Univ. Grenoble Alpes, CNRS, LPMMC, 25 avenue des Martyrs, 38042, Grenoble, France.
**3** Centro de Fisica de Materiales (CFM-MPC), Centro Mixto CSIC-UPV/EHU, Manuel de
Lardizabal 5, E-20018 San Sebastian, Spain and Donostia International Physics Center
(DIPC), Manuel de Lardizabal 4, E-20018 San Sebastian, Spain
**4** Univ. Grenoble Alpes, CNRS, Institut Néel, 25 avenue des Martyrs,
38042, Grenoble, France.
**5** ICTP - Strada Costiera 11, 34151 Trieste, Italy
**6** NEST, Scuola Normale Superiore & Istituto Nanoscienze-CNR, I-56126, Pisa, Italy
**7** Low Temperature Laboratory, Department of Applied Physics,
Aalto University School of Science, P.O. Box 13500, 00076 Aalto, Finland.
**8** Institut für Theoretische Festkörperphysik, Karlsruhe Institute of Technology,
D-76131 Karlsruhe, Germany.

⋆ herve.courtois@univ-grenoble-alpes.fr

## Abstract

**In this article, we provide an overview of the scientific contributions of Frank W. J. Hekking to the fields of mesoscopic electron transport and superconductivity as well as atomic gases. Frank Hekking passed away in May 2017. We hope that the present review gives a faithful testimony of his scientific legacy.**



## Contents

Frank W. J. Hekking performed his PhD work on "Aspects of Electron Transport in Semiconductor Nanostructures" at the TU Delft in 1992. He then worked as a postdoc at the University of Karlsruhe, the University of Minnesota, the Cavendish Laboratory at the University of Cambridge, and the Ruhr University at Bochum. In 1999 he joined the LPMMC (Laboratoire de Physique et Modélisation des Milieux Condensés) in Grenoble and was appointed Professor at the Université Joseph Fourier and afterwards Université Grenoble Alpes. Frank Hekking was nominated as a member of the Institut Universitaire de France, for the periods 2002-2007 and 2012-2017. This review provides an overview of his scientific contributions to several fields of mesoscopic electron transport and superconductivity as well as atomic gases, and is organized along sections describing the different themes.

# 1   One-dimensional conductors

A recurring theme in Frank Hekking's research was electron transport in mesoscopic one-dimensional conductors. The famous formula of Landauer relates the conductance to the quantum mechanical scattering matrix, which in turn depends on the quantum mechanical wave functions of the system. During the early stages of his PhD work in Delft, Frank Hekking and several young experimentalists from Delft and Philips studied the transmission and hence conductance of a fabricated array of quantum dots. They showed that the emerging band structure of this periodic but finite system leads to a series of conductance peaks, see Fig. 1 left. Frank Hekking investigated the eigenenergies of the finite-length chain and the effect of weak disorder. His theoretical predictions for the transmission compared well with the measured values. The results were published in Physical Review Letters [1] and attracted much attention - a good start into a scientific career.

Landauer's formula refers to an ideal situation of quantum mechanical coherent propagation of independent electrons through the structure of interest, e.g., a constriction in a 2-dimensional electron gas between electronic reservoirs. In real systems, the electrons are interacting and are also subject to fluctuations of the electromagnetic environment. To study the environmental effects, Frank Hekking, Yuli Nazarov and Gerd Schön [2, 3] considered a generalization of Landauer's formalism including fluctuations in the transport and gate voltages. The theoretical description is similar to the so-called $P(E)$ theory describing single-electron tunneling in a dissipative environment. The interaction of the electrons with the environment reduces the low-voltage conductance and leads to an offset voltage in the current-voltage characteristic, as shown in Fig. 1 middle, which scales with the external resistance. In the limit of high external resistance, the low-voltage conductance is fully suppressed even for otherwise completely transmitting states.

Interactions introduce qualitatively new features compared to the noninteracting case. In 1-dimensional electronic systems, interactions can be described by the Luttinger model, which predicts rather unconventional behavior such as spin-charge separation. An interesting topic, which Frank Hekking studied together with Rosario Fazio and Arkadi Odintsov [5, 6], are

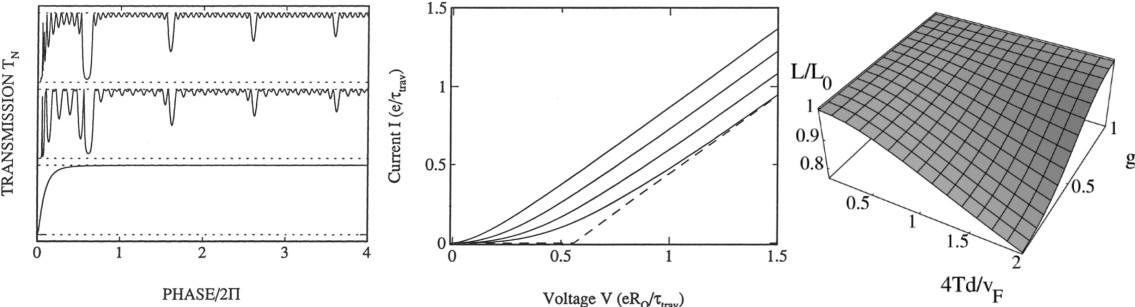

Figure 1: Left: The transmission of an array of 16 quantum dots (upper curve). The transmission of each barrier increases with the phase $\phi$ according to the lowest curve. The middle curve is obtained for a system with weak disorder [4]. Middle: The current (in units of $e/\tau_{\text{trav}}$) through a constriction in series with a resistor $R_s$ as a function of the voltage over the constriction in units of $eR_Q/\tau_{\text{trav}}$ for (from left to right) $R_s/R_Q = 0.5, 1.0, 1.5$, and $2.0$. $\tau_{\text{trav}}$ is the travel time through the constriction. The dashed line is the high-voltage asymptote for $R_s/R_Q = 2.0$ [4]. Right: The Lorentz number for an ideal wire with interaction strength $g$ at temperature $T$ [10].

the properties of superconductor - Luttinger liquid heterostructures. Specifically, they studied the Josephson current through a superconductor - Luttinger liquid - superconductor system separated by tunnel junctions. They found that at zero temperature the Josephson critical current decays algebraically with increasing distance $d$ between the junctions [5] with an exponent which depends on the strength of the interaction. At finite temperatures $T$, there is a crossover from algebraic to exponential decay at a distance of the order of $\hbar v_F/k_B T$. If the Luttinger liquid is confined to a ring of circumference $L$, coupled capacitively to a gate voltage and threaded by a magnetic flux, the Josephson current shows remarkable parity effects upon variation of these parameters. The Josephson current depends on the total number of electrons modulo 4. For some values of the gate voltage and applied flux, the ring acts as a $\pi$-junction. These features are robust against thermal fluctuations up to temperatures on the order of $\hbar v_F/k_B L$.

For the wire geometry, the same authors also studied the ac-Josephson effect [6] and found that the amplitude and phase of the time-dependent Josephson current are affected by the interactions. Specifically, the amplitude shows pronounced oscillations as a function of the bias voltage, which arises because of the difference between the velocities of spin and charge excitations in the Luttinger liquid. Therefore, the ac-Josephson effect can be used as a tool for the observation of spin-charge separation.

Another property of superconductor - Luttinger liquid heterostructures, namely the influence of the superconducting proximity effect on the properties of a Luttinger liquid was studied by Frank Hekking and his coworkers in Ref. [7]. They investigated the frequency and space dependence of the local tunneling density of states (DOS) of a Luttinger liquid (LL) connected to a superconductor. This coupling significantly enhances the DOS near the Fermi level, whereas this quantity vanishes with a power law of this energy difference for an isolated LL. The enhancement is due to the interplay between electron-electron interactions and multiple backscattering processes of low-energy electrons at the interface between the LL and the superconductor. This anomalous behavior extends over large distances from the interface and may be detected by coupling normal probes to the system.

Magnetotunneling provides another method for detecting Luttinger-liquid behavior. Frank Hekking and his coworkers, Alexander Altland, Chris Barnes, and Andy Schofield [8], suggested to measure the tunneling conductance between a quantum wire and a two-dimensional

electron system as a function of both the potential difference V between them and an in-plane magnetic field $B$. They showed that the parameter dependence on $B$ and $V$ allows determining the dependence of the spectral function, $A_{LL}(q, \omega)$ of the quantum wire on the wave vector $q$ and the frequency $\omega$. In particular, the separation of spin and charge in the Luttinger liquid should manifest itself as singularities in the current-voltage characteristics.

In a collaboration with Yaroslav Blanter and Markus Büttiker, Frank Hekking considered a quantum wire coupled to two reservoirs and capacitively coupled to a gate and showed that the strength of the electron-electron interaction can be determined by a low-frequency measurement [9]. Without making use of the Luttinger model, they presented a self-consistent, charge and current conserving theory of the full conductance matrix. The collective excitation spectrum consists of plasma modes with a relaxation rate which increases with the interaction strength and is inversely proportional to the length of the wire.

Frank Hekking also studied thermal transport through quantum wires [10] in a collaboration with Rosario Fazio and Dima Khmelnitsii. They concentrated on smooth structures, with spatial variations on length scales larger than the Fermi wavelength. In this situation, electrons are not backscattered and the electric conductance is not affected, but they found that interactions in the wire still influence thermal transport. Energy is carried by low-energy bosonic excitations (plasmons), which suffer from scattering even on scales much larger than the Fermi wavelength. If the electron density varies randomly, plasmons are localized and a charge-energy separation occurs. As a result they predicted characteristic deviations from the Wiedemann-Franz law depending on temperature and interaction strength, see Fig. 1 right.

More recently, in a collaboration with Michele Filippone and Anna Minguzzi, Frank Hekking studied particle and energy transport for one-dimensional strongly interacting bosons through a ballistic single channel connecting two atomic reservoirs [11]. Again they found the emergence of collective modes with particle- and energy-current separation, and again a violation of the Wiedemann-Franz law. As a consequence, they predicted different time scales for the equilibration of temperature and particle imbalances between the reservoirs. Beyond the linear-spectrum approximation, the thermoelectric effects can be controlled by either tuning interactions or the temperature. The paper provides a unified picture for fermions in condensed-matter devices and bosons in ultracold atom setups.

Together with Leonid Glazman and Anatoli Larkin, Frank Hekking also studied the transport through a quantum dot strongly coupled to reservoirs via a single channel (and hence maximum conductance $e^2/h$) [12]. In the limit of weak conductance, the charge on the dot is quantized, and the system develops, at temperatures below the so-called Kondo temperature, the Kondo effect. For high conductance junctions, the Kondo temperature is increased, which should make it more accessible for experiments, and the charge discreteness is lost. On the other hand, the spin of the dot remains quantized at $s = 1/2$. This spin-charge separation is accompanied by a non-monotonous temperature dependence of the conductance.

## 2 NS junctions, Andreev tunneling

In the early 90's, the physics of hybrid junctions made of a normal metal and a superconductor was believed to be well understood thanks the Blonder-Tinkham-Klapwijk (BTK) [13] model for NIS or NS junctions. The experiment by A. Kastalsky *et al.* [14] demonstrating a differential conductance peak at low bias came as a surprise. The first interpretation of a Josephson coupling between the superconducting lead and the region of the local normal metal with an induced superconductivity by proximity effect does not hold, since only one phase is present in the system. Following the hypothesis of Bart van Wees *et al.* [15], Frank Hekking and Yuli Nazarov calculated the current created by Andreev reflection at the NS interface when the

normal metal is disordered. In the BTK model, the normal metal is considered as ballistic and the probability for Andreev reflection is weak in the case of a tunnel junction. Disorder in the normal metal confines electronic trajectories close to the interface (Fig. 2 top left), making that Andreev reflection probabilities amplitude add up coherently. As a consequence, the conductance of the NIS junction can be written [16, 17]:

$$G_{\text{NIS}} \propto R_{\text{diff}} G_{\text{T}}^2 \,, \tag{1}$$

where $G_{\text{T}}$ is the conductance of the tunnel junction, $R_{\text{diff}}$ is the resistance of the diffusive normal metal. The square exponent on the tunnel conductance expresses the fact that Andreev reflection is a two-particle process. Moreover, the larger the resistance of the diffusive region, the larger the conductance of the junction as a whole. If the superconducting electrode is split into two parts between which a phase difference can be applied, interferences in the Andreev current can be observed [18].

The same phenomenon affects the low-frequency current noise for voltages $V$ and temperatures $T$ much smaller than the superconducting gap. Fabio Pistolesi, Guillaume Bignon and Frank Hekking found that, if the normal metal is at equilibrium, the simple relation

$$S(V, T) = 4e \coth(\frac{eV}{k_{\text{B}}T}) I(V, T) \tag{2}$$

holds quite generally even for nonlinear I-V characteristics [19]. Only when the normal metal is out of equilibrium do noise and current become independent. Their ratio, the Fano factor, depends then on the details of the layout.

Over the years, Frank Hekking investigated the contribution of two-particle tunneling to the properties of hybrid structures. In a superconducting grain connected to two normal electrodes by tunnel junctions, the low-bias conductance is due to electrons passing in pairs through the grain. As a post-doctoral fellow with Leonid Glazman, Frank Hekking showed that the conductance is linear and periodic in the gate voltage [20], see Fig. 2 top right. The period and the conductance activation energy are determined by the charge $2e$ rather than $e$. The competition between charging effects and mesoscopic interference leads to a non-monotonic dependence of the differential subgap conductance on the applied bias voltage. This feature is pronounced, even if the coupling to the environment is weak and the charging energy is small [21]. Two-electron tunneling can also be used for a low-energy spectroscopy of a mesoscopic Josephson junction. Frank Hekking, Leonid Glazman and Gerd Schön showed that the Andreev reflection in the NS junction of a normal-superconductor-superconductor double junction (NSS transistor) provides a unique spectroscopic tool to probe the coherent Cooper-pair tunneling and the energy spectrum of the Josephson (SS) junction [22].

Multi-terminal hybrid structures, consisting of a superconductor with two or more probe electrodes which can be either normal metals or polarized ferromagnets, are very interesting for the creation of electronic quantum entanglement. Besides Andreev pair tunneling at each contact, the subgap transport involves additional channels due to coherent propagation of two particles, each originating from a different probe electrode. The relevant processes are electron cotunneling through the superconductor and conversion into a Cooper pair of two electrons stemming from different probes, see Fig. 2 bottom left. These processes are non-local and appear when the distance between the pair of involved contacts is smaller than or comparable to the superconducting coherence length. Together with Pino Falci and Denis Feinberg, Frank Hekking calculated the conductance matrix of a three-terminal hybrid structure [23]. They observed that multi-probe processes enhance the conductance of each contact. If the contacts are magnetically polarized, the contribution of the various conduction channels can be separately detected. In a following paper with Manuel Houzet and Fabio Pistolesi, Frank Hekking studied the cross-correlation of currents in different leads. By tuning the voltages, it is possible to change the sign of the cross-correlation [24].

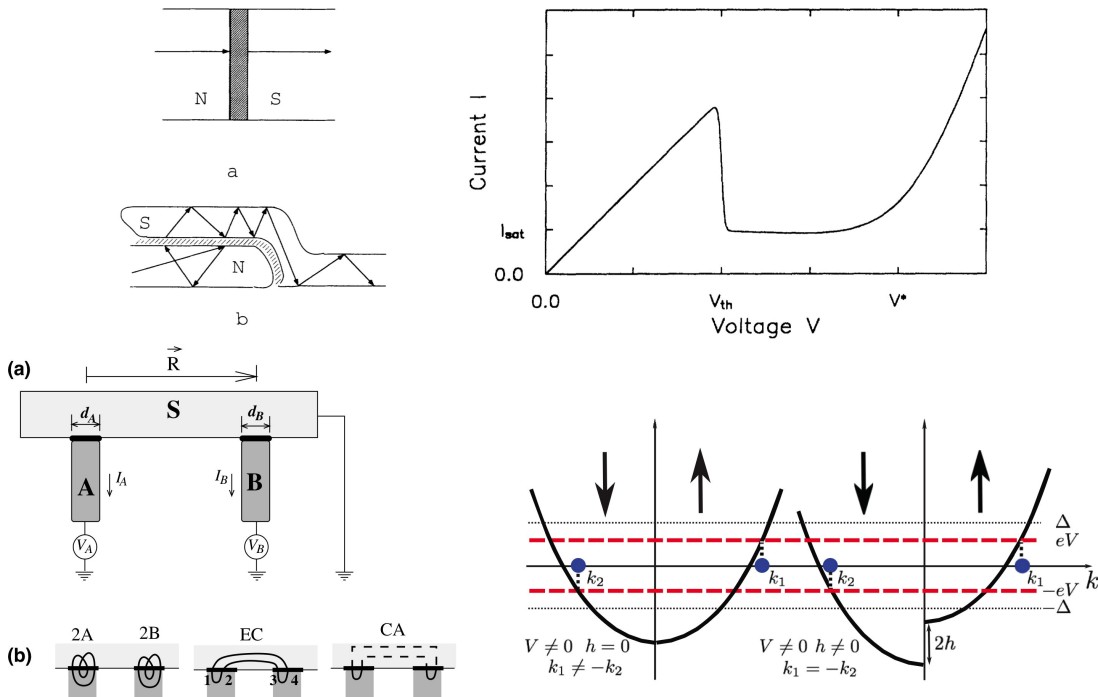

Figure 2: Top left: two realizations of a N-I-S junction, a ballistic junction (top) and an overlap junction where electrons are confined and scatter on the interfaces [16]. Top right: Overall I-V characteristic for a NSNS system with a small superconducting grain [20]. Bottom left: (a) Schematics of the three-terminal A/S/B device; (b) diagrammatic representation of the processes leading to subgap conductance: single-contact two-particle tunneling (2A and 2B), elastic cotunneling (EC), which probes the normal Green's function (full lines) of S and crossed Andreev (CA), which probes the anomalous propagator (dashed line). Bottom right: schematic energy diagram for a non-magnetic (left) and magnetic (right) metal [27]. The thick solid parabolas are the dispersions of free electrons with spin up and spin down. The $k$ axis corresponds to the Fermi level in the superconductor.

Frank Hekking also explored superconducting hybrid structures with ferromagnets (F). In 2004, Yaroslav M. Blanter and Frank Hekking calculated the current-phase relation of a ballistic SFFS Josephson junction [25]. They showed that when the two ferromagnetic layers are of the same size and have equal but opposite magnetization, the amplitude of the supercurrent is independent of the intrinsic exchange field in the F layers. This striking result was explained with the help of a quasiclassical picture, in which a perfect cancelation of the magnetic phase of the electrons takes place while traveling through the ferromagnetic link.

In a later work of Frank Hekking with collaborators, an exhaustive description of the current voltage characteristics of diffusive SFS junctions was provided [26]. Because of the presence of an exchange field, the sign of change on the density of states of the magnetic F layer at the Fermi level depends on the thickness of the F layer. Features of the density of states manifest on the current-voltage characteristics that have been also calculated in Ref. [26].

Another striking effect was predicted in 2012 by Frank Hekking together with Asier Ozaeta, Andrey Vasenko and Sebastian Bergeret [27]. They studied the Andreev current in a SFN junction and found an enhancement of the Andreev current by increasing the intrinsic exchange field of the F interlayer. It is known that the presence of the exchange field is detrimental for superconducting pairing and hence the result of Ref. [27] at first glance was unexpected. Besides

a quantitative analysis of this effect, Frank Hekking and collaborators provided a physical interpretation (see Fig. 2 bottom right) based on Hekking's work on the two-electron tunnelling processes that give rise to the subgap current [16, 17]: Two-electron tunnelling is a coherent process and its main contribution stems from two nearly time-reversed electrons $k_1 \simeq k_2$ located at the energy window $\delta\epsilon \sim eV, T$ around the Fermi level [28], defined by the applied voltage and the temperature (*cf.* Fig. 2 bottom right). In the presence of an exchange field $h$, majority and minority spin electrons at the Fermi level are characterised by different momenta $k_{1,2} \simeq k_F \pm h/v_F$. As shown in Fig. 2 bottom right, the wave vectors $k_{1,2}$ are determined by the intersection between the parabolas and the $k$ axis. In the absence of $h$, *i.e.* for a non-magnetic metal, the relevant excitations with energies $\sim \pm eV$ are not time-reversed (see left panel of Fig. 2 bottom) and therefore do not contribute to the Andreev current. However, by increasing $h$, $|k_1| \to |k_2|$ the relevant excitations become more and more coherent, leading to the possible increase of the Andreev current by increasing $h$ [27].

## 3 Superconducting fluctuations, granular metals

In samples with reduced dimensions, deviations from the mean-field BCS behaviour can be sizeable. An important example of this kind are the precursors of the superconducting instability in the conductivity, the Aslamazov-Larkin and Maki-Thompson contributions. Fluctuation corrections to equilibrium and transport quantities have been extensively studied since the early days of superconductivity and a detailed account of this activity can be found in the book of Larkin and Varlamov [29]. Several works of Frank Hekking's activity were concerned with the study of fluctuations in superconducting systems. They focused in particular on the study of superconducting loops, granular materials and small superconducting particles.

Frank Hekking and collaborators highlighted a number of interesting effects induced by (classical or quantum) superconducting fluctuations. Together with Leonid Glazman and Alexander Yu. Zyuzin, Frank Hekking showed that the electrical response of a superconducting loop close to the transition is highly non-local due to the diverging correlation length on approaching the transition point [30]. As a result of this, it was shown that rings with a radius of the order of the correlation length would be sufficient to detect the effect. The set-up that they consider is shown in Fig. 3 left. Using a time-dependent Ginzburg-Landau theory, they calculated the flux dependence of the voltages $V_\alpha$ and $V_\beta$ as a function of a magnetic field piercing the loop. In addition to the standard (local) Aslamazov-Larkin contribution, a new non-local term appears. It manifests itself in oscillations of the ratio $V_\alpha/V_\beta$ as a function of the flux. In the geometry of Fig. 3, the result is:

$$\frac{V_\alpha}{V_\beta} - \frac{1}{3} \sim \frac{R}{\xi(0)} \sqrt{\frac{T_c}{T - T_c}} e^{-2\pi R/\xi(T)} \cos 2\pi \frac{\Phi}{\Phi_0} \,, \tag{3}$$

where the $T_c$ is the mean-field transition temperature, $\Phi_0$ is the flux quantum and $\Phi$ the applied flux.

Fluctuations effects in superconducting loops were also analysed by Frank Hekking and Leonid Glazman by looking at their effects on the supercurrent [31]. In this work, they mainly concentrated on how quantum fluctuations renormalise the persistent current. In a short loop, the current as a function of the applied flux is still sinusoidal, with a suppressed amplitude. On the other hand, in a large loop, when the supercurrent shows a saw-tooth dependence in the classical limit, they observed a smearing of the cusps of the saw-tooth dependence due to the presence of quantum fluctuations. The non-trivial behaviour of quantum corrections manifests in the dependence of the amplitude of the current that shows a power-law dependence on the junction conductance, with an exponent depending on the low-frequency impedance of the

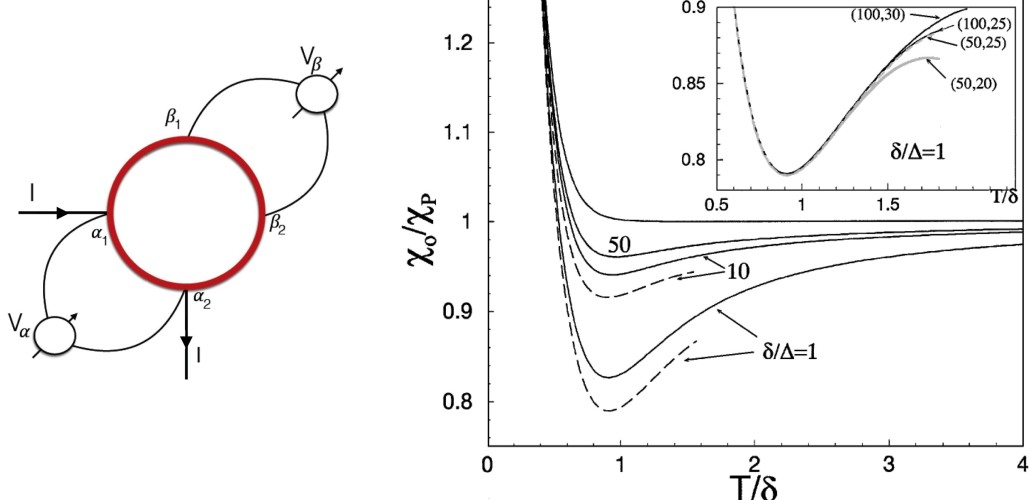

Figure 3: Left: In the four-terminal setup considered in Ref. [30] the two current probes are attached in $\alpha_1$ and $\alpha_2$ while the voltage probes are in $\beta_1$ and $\beta_2$. The dimensions of the ring (in red) should be of the order of the superconducting length $\xi$. Right [35]: The susceptibility of a small grain with a odd number of electrons is shown as a function of temperature, parametrised by different values of the ratio $\delta/\Delta$ ($\delta$ is the level spacing and $\Delta$ is the BCS superconducting gap). This ratio quantifies the size of superconducting fluctuations. On increasing the pairing fluctuations, the non-monotonous behaviour becomes more pronounced. The inset shows details of the approximations used in the numerical evaluation of the pairing fluctuations.

ring.

The interest of Frank Hekking in granular systems spanned over several different aspects. In [32], together with Cristiano Biagini, Raffaello Ferone, Rosario Fazio and Valerio Tognetti, the superconducting fluctuation corrections to the heat conductivity were considered. A strong singular correction is found, due to both the renormalization of the density of states and the Aslamazov-Larkin contributions. Granular systems were further investigated in [33, 34] in joint works with Igor S. Beloborodov, Andrei V. Lopatin, Rosario Fazio, and Valery M. Vinokur and with Alexander Altland, Igor S. Beloborodov, and Konstantin B. Efetov. Thermal transport was also central to the work in [33] where it was shown that at intermediate temperatures interactions lead to a violation of the Wiedeman-Franz law in a granular metal. In Ref. [34], Frank Hekking and co-workers showed that in order to describe correctly the properties of granular metals, the diffusive nature of the electronic motion within the grains is crucial. They discussed these effects using both a diagrammatic and a functional integral approach deriving a new effective action to describe the array at arbitrary temperature. While at high temperature the new effective action reduces to the Ambegaokar-Eckern-Schön form, in the low-temperature limit their analysis yields the dynamically screened Coulomb interaction of a normal metal.

Finally, with the group in Catania (Antonio Di Lorenzo, Pino Falci, Rosario Fazio, Gaetano Giaquinta, and Andrea Mastellone) superconducting fluctuations in small grains were studied in [35] through an analysis of the temperature dependence of the spin susceptibility. In a bulk superconductor, the spin susceptibility $\chi$ decays to zero exponentially due to the presence of the pairing gap. In a small grain, the situation changes dramatically due to the interplay between parity effects and pairing correlations. In particular, if the number of electrons on the grain is odd, a re-entrant behaviour is observed as a function of temperature. This is shown in

Fig. 3 right where the susceptibility, in units of the bulk high-temperature value, is shown as a function of the temperature (in units of the level spacing $\delta$). The interest of Frank Hekking in studying the properties of superconductivity in small grains was not confined to understanding how to reveal pairing fluctuations in finite-size systems. His attention was devoted to general properties of pairing in small systems ranging from grains to atomic nuclei. In collaboration with Michel Farine, Peter Schuck and Xavier Vinas [36], he showed that the presence of a surface may be responsible for the enhancement of pair coupling.

This section devoted to Frank Hekking's work on fluctuations can be completed mentioning his results, obtained in collaboration with Fabio Taddei [37], on a proposal to probe the distribution of current fluctuations by means of the escape probability histogram of a pulsed Josephson junction. With this scheme, they were able to extract information on the moments of the distribution of current, a problem of great relevance in connection with the study of counting statistics and, more recently, of work statistics. Their method turned out to be quite general and could be applied to a variety of different situations.

## 4  Adiabatic pumping, environment, quantum thermodynamics

One of Frank Hekking's interests was adiabatic pumping of charge in quantum circuits. Due to the interaction with the experimental activities at Aalto University in Helsinki, the work focused on Josephson junction based quantum pumps. On this topic Frank Hekking worked in close collaboration with Rosario Fazio and Jukka Pekola.

The charge can be pumped unidirectionally with two or more gate voltages or magnetic fluxes applied in a cyclic manner. The devices where this effect has been investigated are a multi-junction Cooper-pair array and a Cooper-pair sluice [38], a charge pump where Cooper-pairs are transported coherently by radio-frequency control realized by combined magnetic fluxes and electric field (gate voltage). The interesting effects in adiabatic pumping relate to its geometric nature, in particular to its relation to the Berry phase, and to the influence of dissipation and measurement on the pumped current.

The majority of works consider the limit where the Josephson coupling energy $E_J$ between the various parts of the system is small compared to the Coulomb charging energy $E_C$. In this case, the charge $Q_P$ transferred in a pumping cycle is about $2e$, the charge of one Cooper pair: the main contribution is due to incoherent Cooper pair tunneling. Yet there is something called "quantum correction" to $Q_P$, which is due to coherent tunneling of pairs across the pump and which depends on the superconducting phase difference $\phi$ between the electrodes: $Q_P/(2e) = 1 - (E_J/E_C)\cos\phi$. It is fair to say that preserving the coherent contribution may be interesting from a fundamental point of view, but still the main aim of pumping experiments is to achieve a metrologically accurate device transferring precisely an integer number of electrons or Cooper pairs in each cycle. From this point of view, a Cooper-pair pump with coherent transfer of charge is not ideal, at least in the configurations investigated in the works presented here. In Ref. [39] the influence of current measurement on pumping was investigated for the first time. A measurement of $Q_P$ tends to destroy the phase coherence. An arbitrary measuring circuit was first studied, and then specific examples which demonstrated that coherent Cooper pair transfer can in principle be detected using an inductively shunted ammeter.

In the work by Frank Hekking and colleagues from Helsinki and Pisa [40], a measurement scheme for observing the Berry phase in a flux assisted Cooper pair pump (Cooper pair sluice) was presented. In contrast to common experiments, in which the pump generates current through a resistance, a device in a superconducting loop was considered. This arrangement introduces a connection between the pumped current and the Berry phase accumulated during adiabatic pumping cycles. From the adiabaticity criterion, equations for the maximum pumped

current were obtained and they allowed to optimize the sluice accordingly. These results were also considered in view of sluice being a potential candidate for a metrological current standard. For measuring the pumped current, an additional Josephson junction was proposed to be installed into the superconducting loop. It was shown in detail that the switching of this system from superconducting state into normal state as a consequence of an external current pulse through it may be employed to probe the pumped current. This scheme was realized soon after by the Aalto group, and the experiment (Fig. 4 top) served as a demonstration of Berry phase in a Josephson junction circuit [41].

In collaboration with the Pisa Group, Frank Hekking developed a formalism to study adiabatic pumping through a superconductor-normal-superconductor weak link. At zero temperature, the pumped charge is related to the Berry phase accumulated, in a pumping cycle, by the Andreev bound states. The pumped charge turns out to be an even function of the superconducting phase difference [42]. With the same group and Alain Joye at Institut Fourier in Grenoble, Cooper pair pumping was considered as a coherent process and a general expression was discussed for the adiabatic pumped charge in superconducting nanocircuits in the presence of level degeneracy. In a proposed experimental system, the non-Abelian structure of the adiabatic evolution manifests in the pumped charge [43].

A recurring topic in Frank Hekking's research was dissipation and the influence of the environment on mesoscopic transport of charge and heat. One of these works was related to electronic cooling described elsewhere in this review. In the work by Frank Hekking and Jukka Pekola [44], and further in [45] with other colleagues from Helsinki, a phenomenon of cooling by heating was explored, in a configuration where a normal metal - superconductor tunnel junction acts as a Brownian refrigerator. Thermal noise generated by a hot resistor (resistance $R$) can, under proper conditions, promote heat removal from a cold normal metal (N) in contact with a superconductor (S) via a tunnel barrier (I). Such a NIS junction acts as a kind of a Maxwell demon, promoting heat flow from a cold reservoir into a hot one. Upon reversal of the temperature gradient between the resistor and the junction, the heat fluxes are reversed: this presents a regime which is not accessible in an ordinary voltage-biased NIS structure. Analytical results for the cooling performance in an idealized high impedance environment were obtained in [44], and numerical ones for general $R$. The system appears experimentally feasible. Explicit analytic calculations showed that the entropy of the total system is always increasing [45]. Further a single electron transistor configuration with two hybrid junctions in series was considered, and it was demonstrated how the cooling is influenced by charging effects. The cooling effect from nonequilibrium fluctuations instead of thermal noise was also demonstrated, focusing on the shot noise generated in another tunnel junction.

Again in collaboration with experimentalists, Frank Hekking and his student Angelo Di Marco investigated theoretically how photon-assisted tunneling processes due to the presence of a high-temperature environment influence the accuracy of a hybrid single-electron turnstile [46]. This device consists of a gate-controlled normal-metal island connected to two voltage-biased superconducting leads by means of two tunnel junctions. Photon assisted tunneling is one of the error sources in the precise transfer of electrons for metrological purposes. In the first work of Angelo Di Marco, Ville Maisi together with their supervisors from Grenoble and Helsinki [47], a voltage-biased NIS tunnel junction, connected to a high-temperature external electromagnetic environment was considered. This model system features the commonly observed subgap leakage current in NIS junctions through photon-assisted tunneling which is detrimental for applications. The subgap leakage current was studied both analytically and numerically; the link with the phenomenological Dynes parameter was discussed. An improved circuit with a reduced influence of photon assisted tunneling is obtained if a low temperature lossy transmission line is inserted between the NIS junction and the environment. It was shown that the subgap leakage current is exponentially suppressed as the length, and

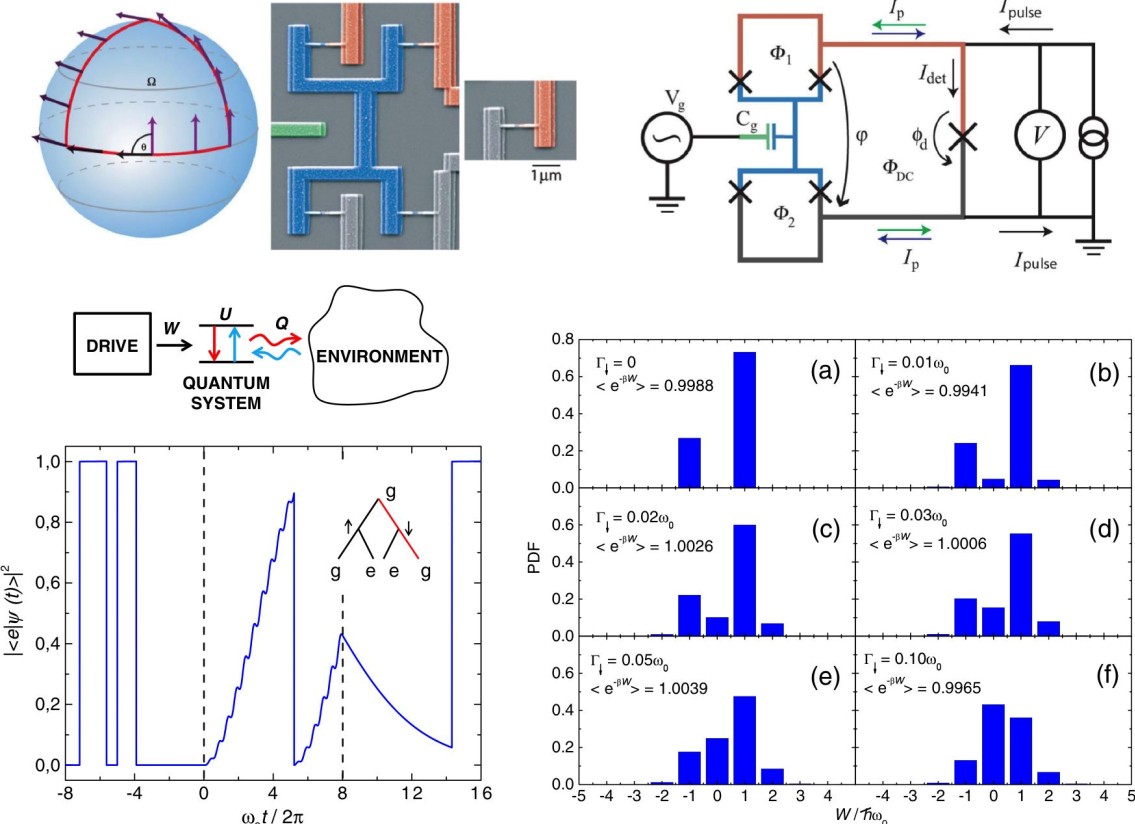

Figure 4: Top, from left to right [41]: Parallel transport of a vector along a path (red line) enclosing a solid angle $\Omega$. The light arrow shows the initial state of the vector and the dark arrow the final state. The angle between the initial and the final vectors $\theta$ is equal to $\Omega$. Scanning electron micrograph of the actual structure measured: island on the left and the detector on the right. Simplified circuit diagram of the measured sample. The corresponding parts in the circuit diagram and SEM-images are marked by colors. Middle: Quantum jump approach to study stochastic thermodynamics of an open quantum system [51]. A schematic representation of an open quantum two-level system exchanging energy via photon emission/absorption to/from the heat bath. Bottom left: a typical time evolution of the stochastic wave function (population of the excited state) when the qubit is driven. Quantum jumps are seen as abrupt changes of the population. Bottom right: extracted work distributions of the driven qubit based on the quantum jump approach. Different panel correspond to different decay rates of the qubit. In all cases the central non-equilibrium fluctuation relation (Jarzynski equality) is satisfied within the numerical accuracy.

the resistance per unit length of the line are increased.

Beyond sequential tunneling, the effect of photon-assisted Andreev reflection on the accuracy of the turnstile was investigated in Ref. [48]. The exchange of photons between the system and the high-temperature electromagnetic environment enhances Andreev reflection, thereby limiting the single-electron tunneling accuracy. Finally, concerning noise and dissipation, Frank Hekking interacted with colleagues in Helsinki during his sabbatical stay in 2005 studying fluctuations at finite frequencies using scattering theory [49,50].

Relatively recently, Frank Hekking studied problems in quantum thermodynamics. In particular, he was able to address the question of how heat is transported between a quantum system and environment via photon emission and absorption. In a joint work of Frank Hekking

and Jukka Pekola [51], the well-known method of quantum jumps [52] was harnessed to address the energetics of open quantum systems, see Fig. 4 bottom.

In Ref. [51] this approach was applied to find the statistics of work in a driven two-level system coupled to a heat bath. It was found that the common non-equilibrium fluctuation relations are satisfied identically using this method. The usual fluctuation-dissipation theorem for linear response was demonstrated for weak dissipation and/or weak drive. Qualitative differences between the classical and quantum regimes, in particular the structure of quantum trajectories as opposed to alternating classical trajectories was described in this work. This work serves as a stepping stone towards thermodynamics experiments in superconducting quantum circuits, where measurement of photons by calorimetric means seems feasible [53]. It is regrettable that Frank Hekking could not continue this work towards experimentally feasible systems and protocols.

## 5 Quantum dynamics in superconducting circuits

As he arrived in Grenoble in 1999, Frank Hekking widened his research activity to the emergent field of superconducting qubits in a strong collaboration with Olivier Buisson at CRTBT (now Néel Institute). Their first motivation was to consider a superconducting quantum circuit which can reproduce Quantum Electrodynamics (QED) physics e.g., a Rydberg atom in high-Q cavity [54]. In a first proposal, a charge qubit capacitively coupled to a LC resonator was considered [55], see Fig. 5 top left. In such a circuit the charge qubit corresponds to the Rydberg atom and the LC resonator to the high-Q cavity. The theoretical study predicted a strong coupling between the charge qubit and the resonator with a coupling strength in the range of 100 MHz. This strength was three orders of magnitude larger than the coupling between a Rydberg atom and a high-Q cavity. To our knowledge, this was the first theoretical prediction on circuit QED experiments [56].

From this initial circuit, Frank Hekking rapidly moved to considering a Josephson junction coupled to a Cooper-pair box. In this circuit he pointed out the interest to work at the degeneracy point of the charge qubit in order to weaken the charge noise effect on the qubit decoherence [57]. By current-biasing the Josephson junction, a quantum measurement of the charge qubit was proposed, based on vacuum Rabi oscillation between a Cooper pair box and a Josephson junction and switching detection [58]. This theoretical proposition was experimentally demonstrated some years later during Aurélien Fay's PhD thesis when Wiebke Guichard joined the Grenoble team. In this experiment, the time domain quantum dynamics in a circuit was studied by coupling an asymmetric Cooper pair transistor to a dc SQUID [59].

Frank Hekking was motivated to derive the full Hamiltonian of the generic circuit based on a Cooper pair transistor coupled to a current-biased Josephson junction or dc SQUID, see Fig. 5 bottom left. In collaboration with Wiebke Guichard, he performed an extensive theoretical analysis [60] and showed that the circuit can be modeled as a charge qubit coupled to an anharmonic oscillator (dc SQUID). Depending on the anharmonicity of the dc SQUID, the Hamiltonian can be reduced either to one that describes two coupled qubits or to the Jaynes-Cummings Hamiltonian. The coupling term, which is a combination of a capacitive and a Josephson coupling between the two qubits, can be tuned from the very strong- to the zero-coupling regime. It describes very precisely the tunable coupling strength measured in this circuit as well as the adiabatic quantum transfer readout [59]. In addition, this theoretical study explains some observations of the quantronium experiment [61] in which a Cooper pair transistor is coupled to a biased Josephson junction.

In addition to these two coupled quantum systems, Frank Hekking and Olivier Buisson considered the quantum dynamics in a current biased dc SQUID. This quantum system is de-

scribed by a driven anharmonic oscillator. At low microwave power, the dc SQUID corresponds to a phase qubit with only the two first levels involved in the quantum dynamics. This limit is relevant when a zero-biased SQUID with a quartic potential was studied and a Camelback phase qubit demonstrated [62], results obtained in collaboration with Emile Hoskinson and Florent Lecocq at Néel Institute.

At higher microwave drive amplitude, more levels are involved in the dynamics of the anharmonic oscillator. During Julien Claudon's PhD thesis, Frank Hekking and collaborators discussed this multilevel time domain quantum dynamics as well as the cross-over between the two-level and multilevel regimes [63, 64], see Fig. 5 top right. Theoretical calculations performed by Frank Hekking could describe precisely this multilevel dynamics and predict several Rabi oscillations frequencies and beating effects between them. In this higher drive amplitude limit, the two-level system has to be extended to a multi-level quantum system. In such a complex system, in a collaboration with Hamza Jirari, an optimal control scheme was theoretically proposed to reach the quantum state target [65]. Related to multilevel dynamics, Frank Hekking together with Yaroslav M. Blanter and Nicolas Didier have considered the dynamics of an open quantum system consisting of a three-level system coupled to a cavity and to a strong external drive [66]. This theoretical analysis described the lasing effect observed in a superconducting nanocircuit where a Cooper-pair box, acting as an artificial three-level atom, was coupled to a resonator [67].

In addition, by comparing quantum and classical dynamics, Frank Hekking in collaboration with Alex Zazunov and the experimental group at Néel Institute showed that Rabi-like oscillations exist both in the quantum and classical regime. However, the Rabi frequency dependence on the microwave amplitude presents a clear quantum signature when up to three levels are involved. At higher excitation amplitude, classical and quantum predictions for the Rabi frequency converge even in absence of decoherence processes. This result has been discussed in the light of a calculation of the Wigner function which points out that pronounced quantum interferences always appear in the course of the Rabi oscillations even at higher excitation amplitude [63].

In all these experiments, quantum measurements were based on escape or switching processes from a metastable state of a dc SQUID [68]. Quantum measurements in the Camelback phase qubit constitute a novel macroscopic quantum tunneling effect. Indeed in such a quantum system the potential is quartic, instead of the usual cubic shape. This led to a double escape path effect as demonstrated in the experiment and theoretically analysed in Nicolas Didier's PhD thesis [62, 69].

Somewhat related to the experiments on adiabatic pumping, Frank Hekking's interest was directed to dissipation in Josephson junctions with relatively small $E_J$. Of particular interest was the experimental observation by Jani Kivioja *et al.* [70] of the transition from escape dynamics of a Josephson junction to underdamped phase diffusion. The usual crossover of underdamped Josephson junctions occurs between macroscopic quantum tunnelling and thermally activated (TA) behaviour upon increasing the temperature. In the work of Jani Kivioja *et al.*, the transition from TA behaviour to underdamped phase diffusion was observed experimentally on increasing the temperature. Above the crossover temperature, the threshold for switching into the finite voltage state becomes extremely sharp, in strong contrast to the broadening of the threshold in the TA regime. Frank Hekking, Jukka Pekola, Olivier Buisson and collaborators [70, 71] proposed a phase-diagram in the $(T, E_J)$ plane in various regimes (Fig. 5 bottom right), and showed that dissipation and level quantization in a metastable well are important ingredients. Two other independent experiments demonstrated a very similar transition at about the same time [72, 73].

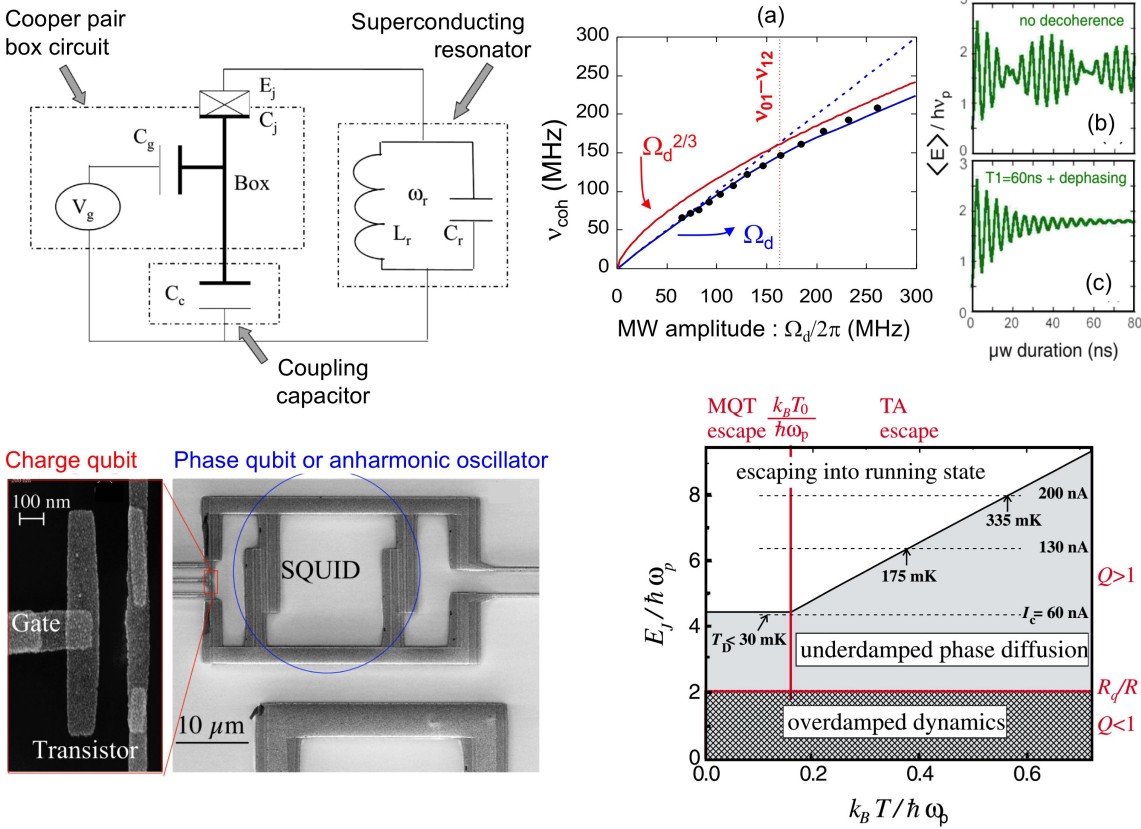

Figure 5: Top left: Quantum superconducting circuits inspired from a Rydberg atom in a high Q-cavity. The Cooper pair box corresponds to the Rydberg atom, the resonator to the high Q cavity and the coupling is realized by a capacitor [55]. Top right: a) Rabi-like oscillation frequency as function of the microwave drive amplitude. The cross-over between the two-level limit and multilevel dynamics is indicated by the vertical dashed line. The experimental data is fitted by the quantum model (blue line) and is compared to a classical model (red line). b) and c) Average energy Rabi-like oscillations predicted by the quantum model in absence and presence of decoherence, respectively [63]. In absence of decoherence, a beating effect was predicted in the Rabi-like oscillations. Bottom left: The experimental implementation of the coupled circuit. The charge qubit is realized by an asymmetric Cooper pair transistor and the resonator by a dc SQUID which is described by an anharmonic oscillator [59]. Bottom right: Phase diagram in the different dynamics of a current biased Josephson junction with low $E_J$ as function of normalized temperature and Josephson energy [70].

## 6 Electronic cooling

Electronic cooling occurs in a Normal-metal - Insulator - Superconductor (NIS) junction when biased at a voltage just below the energy gap $\Delta/e$ [74]. In that case, only high-energy electrons can leave the normal metal, or alternatively depending on the bias sign, only low energy electrons can enter. Actual devices are made of a pair of NIS junctions, arranged in a SINIS geometry. In that case, the cooling power is doubled and the normal metal is thermally well isolated. As the electron flow through the tunnel barriers is weak, a quasi-equilibrium situation is reached in the normal metal: the electronic energy distribution function follows a Fermi-

Dirac statistics at a well-defined *electronic* temperature $T_e$. The latter can easily be determined by another NIS junction biased at a current low enough for the associated energy flow to be negligible. The cooling power of the NIS junctions can be written as:

$$\dot{Q}_N^0 = \frac{1}{e^2 R_N} \int_{-\infty}^{\infty} (E - eV) n_S(E)[f_N(E - eV) - f_S(E)]dE, \tag{4}$$

where $R_N$ is the normal state resistance, $f_{N,S}$ the electron energy distribution in the normal metal or the superconductor and $n_S(E)$ the normalized BCS density of states in the superconductor. In a heat balance approach, it is counter-balanced by the electron-phonon coupling power

$$P_{e-ph}(T_e, T_{ph}) = \Sigma U(T_e^5 - T_{ph}^5), \tag{5}$$

where $T_{ph}$ is the phonon temperature, $\Sigma$ is a material-dependent parameter and $U$ is the volume. Early experiments have demonstrated the electronic cooling effect, in qualitative agreement with predictions. Still, the observed cooling remained lower than expected.

This puzzle was addressed in a joint study between the group of Jukka Pekola at Helsinki, Francesco Giazotto being a visitor there at that time, and Frank Hekking [75], see Fig. 6 top left. The experimental cooling curve displaying the electronic temperature as a function of the cooler bias strikingly showed a non-monotonous behavior. Both an out-of-(quasi)equilibrium energy distribution and a non-zero density of states within the superconductor energy gap were invoked to explain the deficit of electronic cooling as compared to theoretical predictions. The latter was shown to determine a lower limit in the electronic temperature.

The limitations in electronic cooling were further studied in the thesis work of Sukumar Rajauria in the group of Bernard Pannetier and Hervé Courtois at CRTBT (now Néel Institute). The current-voltage characteristics of SINIS junctions was measured with a high accuracy [76]. A logaritmic scale plot of the *I-V* reveals two different regimes for low and high bias, see Fig. 6 bottom right. Frank Hekking and Sukumar Rajauria calculated the effect of Andreev reflection on the heat current. While the Andreev charge current can usually be safely neglected in junctions with rather opaque barriers, the related heat current brings in many cases a significant contribution. This is because the single-electron tunneling current has a cooling efficiency $k_B T_e/\Delta$ of about a few percent, while the Andreev current generates Joule heat with a 100 % efficiency. The experimental data were very well fitted using a heat balance taking into account the Andreev current [77]. The electronic temperature calculated with the sample parameters shows a non-monotonous dependence with the cooler bias. While the latter theoretical calculations were carried out using a tunnel Hamiltonian approach, a later work in collaboration with Andrei Vasenko demonstrated that the same physics could be studied using the Usadel equations formalism [78].

The coupling to phonons was a major issue in the analysis of electron cooling, in particular the usual assumption that phonons are well thermalized to the bath. In collaboration with the CRTBT group, Frank Hekking demonstrated that the thermal behavior of coolers, especially in the intermediate temperature range, could be understood by assuming a decoupling of the cooled metal phonons from the substrate phonons [79]. This hypothesis was later confirmed through the independent measurement of the phonon temperature in Laetitia Pascal's PhD thesis [80] and the enhanced cooling capabilities in large-power coolers [81]. In a narrow suspended metallic wire, in which the phonon modes are restricted to one dimension but the electrons behave three-dimensionally, longitudinal vibration modes can be cooled by electronic tunneling. The refrigeration can reach temperatures far below the bath temperature provided the mechanical quality factors of the modes are sufficiently high [82], see Fig. 6 top right.

In electronic cooling, hot quasi-particles are extracted from a normal metal and hence injected in a superconductor. Their evacuation is usually ensured by normal metal traps placed

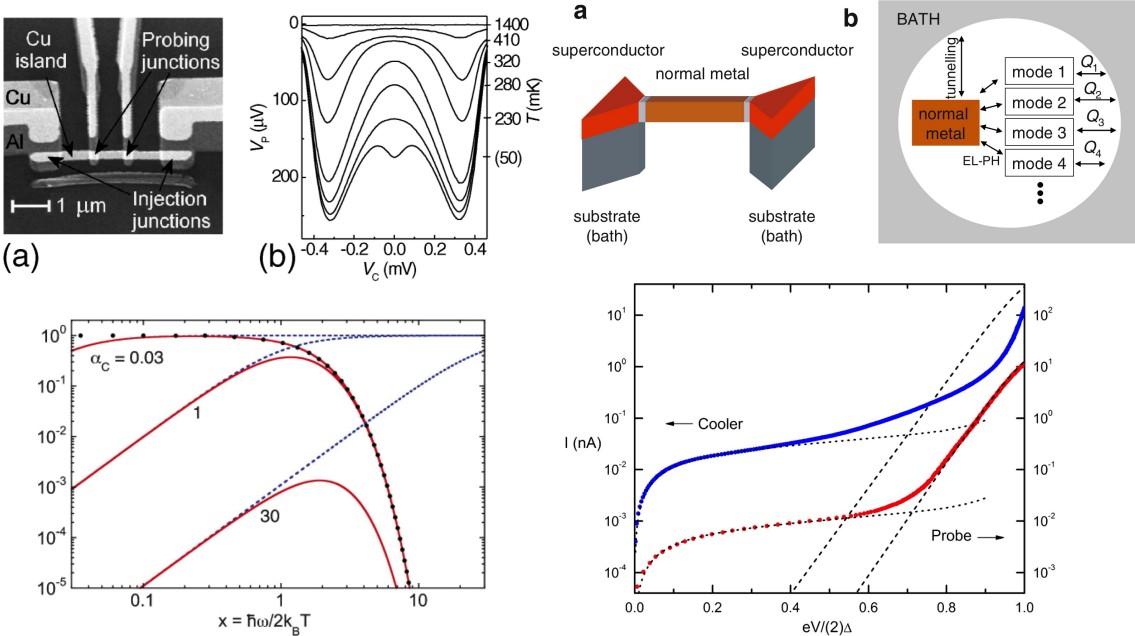

Figure 6: Top left: Picture of a typical cooler (a), and cooling data (b) where the voltage $V_P$ across the probe junctions biased with a constant current is shown against the voltage $V_C$ across the two injection junctions. The electron temperature on the N island at $V_C = 0$ is indicated on the right vertical axis [75]. Top right: A suspended wire connected to superconducting reservoirs via tunnel barriers to form an electronic cooler, and the relevant thermal model. The transverse dimensions are assumed to be smaller than the thermal phonon wavelength [82]. Bottom left: Photonic channel for heat in the case of a capacitive coupling. Spectrum of the thermal noise power density (black dots), the photon transmission coefficient (dotted blue line), and of photonic heat (thin red full line) as a function of the frequency for values of the capacitance. The frequency is plotted in units of the thermal frequency $2k_B T/\hbar$. We consider the case of perfect resistance matching $R_1 = R_2 = R$. Bottom right: Current-voltage characteristic of the cooler junctions (top curve) and of the probe (bottom curve) as a function of the voltage together with calculated Andreev current (dotted lines) and single-particle current (dashed lines) [76].

in the vicinity of the junction. Still, their efficiency is limited by the (usually) weak transparency of their interface with the overheated superconducting lead, as was studied in a joint experimental-theoretical work involving Frank Hekking and the CRTBT group [83]. Eventually a cascade geometry based on a combination of superconducting tunnel junctions was theoretically investigated in collaboration with the group of Francesco Giazotto in Pisa, and the group of Hervé Courtois and Clemens Winkelmann at Néel Institute [84]. The cascade extraction of hot quasiparticles, which stems from the energy gaps of two well-chosen different superconductors, allows for a normal metal to be cooled down to about 100 mK starting from a bath temperature of 0.5 K.

Another thermal channel coupling an electronic bath to the external world is photonic. The heat transfer by photons between two metals can be described as a circuit containing linear reactive impedances. Using a simple circuit approach, Frank Hekking, Laetitia Pascal and Hervé Courtois have calculated the spectral power transmitted from one metal to the other and found that it is determined by a photon transmission coefficient which depends on the impedances of the metals and of the coupling circuit [85], see Fig. 6 bottom left. The

total photonic power flow was studied for different coupling impedances both in the linear regime where the temperature difference between the metals is small and in the nonlinear regime of large temperature differences. Frank Hekking and Hervé Courtois continued to work afterwards on the topic of radiative heat transport at mesoscopic scale, in particular the question of the crossover between the near-field and the far-field regime.

Besides the electron cooling in SIN structures, Frank Hekking also studied the cooling effect in superconducting hybrid structures containing ferromagnetic materials [86, 87]. In particular, in Ref. [86] it was demonstrated that by using a spin-filter tunneling barrier (e.g. a ferromagnetic insulator) the extraction of heat from a normal metal can be more efficient than by using non-magnetic tunneling barriers.

# 7   Josephson junction chains

Frank Hekking's activity on Josephson junction chains came as a natural extension of his work on SQUIDs and qubits (described in Sec. 5 above), and was marked by a close collaboration with Wiebke Guichard. The degree of control in the sample fabrication process reached by the end of the decade 2000-2010 suggested Josephson junction-based circuits as a promising platform for building various devices [88, 89]. Already in 2006, Frank Hekking provided theoretical support for experiments performed by David Haviland and his group in Stockholm including at that time Wiebke Guichard, where Josephson junction chains were used to create a controllable electromagnetic environment [90, 91].

In 2006, Hans Mooij and Yuli Nazarov put forward the idea of a new device, the quantum phase-slip junction, which, when irradiated by microwaves, could serve as the implementation of a new current standard based on phase-charge duality [92]. The central element of that proposal was a thin superconducting wire where coherent quantum phase slips (QPSs) would occur. In 2010, Frank Hekking together with Wiebke Guichard showed that the same idea could also work if one uses a Josephson junction or a chain of Josephson junctions as the QPS element, instead of a wire [93]. Such implementation might prove more advantageous because fabrication of high-quality thin wires represents a greater challenge from the technological point of view. A few years later, Frank Hekking, Gianluca Rastelli and Angelo Di Marco performed a detailed theoretical study of the QPS element subject to a microwave radiation and elucidated the crucial role of the environment in such a device [94]. The effect of the environment on a QPS element was studied in an experiment by the group at Néel Institute [95], where the environment was represented by a Josephson junction chain, and the dc current-voltage characteristics of the device [Fig. 7 (a)] were interpreted in terms of the dynamics of the charge degree of freedom, dual to the phase, which arises as a result of coherent QPSs.

The potential use of Josephson junction chains as implementations of a current standard necessitated a better theoretical understanding of the coherent QPSs in such chains. At that time, the standard theory of QPSs in Josephson junction chains was the one due to K. Matveev *et al.* [96]. In that work, the only effect of the Coulomb interaction included was the electrostatic coupling via a capacitance $C$ between the neighboring islands forming each Josephson junction [Fig. 7(b)], while the self-capacitance $C_0$ of the islands was neglected. In such a model, the QPS on each junction does not disturb other junctions; as a result, the total coherent QPS amplitude $W$ for the chain has a trivial dependence on the junction number $N$: $W \propto N$, or $W \propto \sqrt{N}$ if random offset charges are present which give random phases to the QPS on different junctions. In collaboration with Ioan M. Pop and Gianluca Rastelli, Frank Hekking led the theoretical effort to calculate the QPS amplitude taking into account the island self-capacitance [97]. It was shown that even a small $C_0 \ll C$ produces a dramatic effect for sufficiently long chains, $N \gtrsim \ell_s \equiv \sqrt{C/C_0} \gg 1$: the $N$ dependence of the QPS amplitude $W$

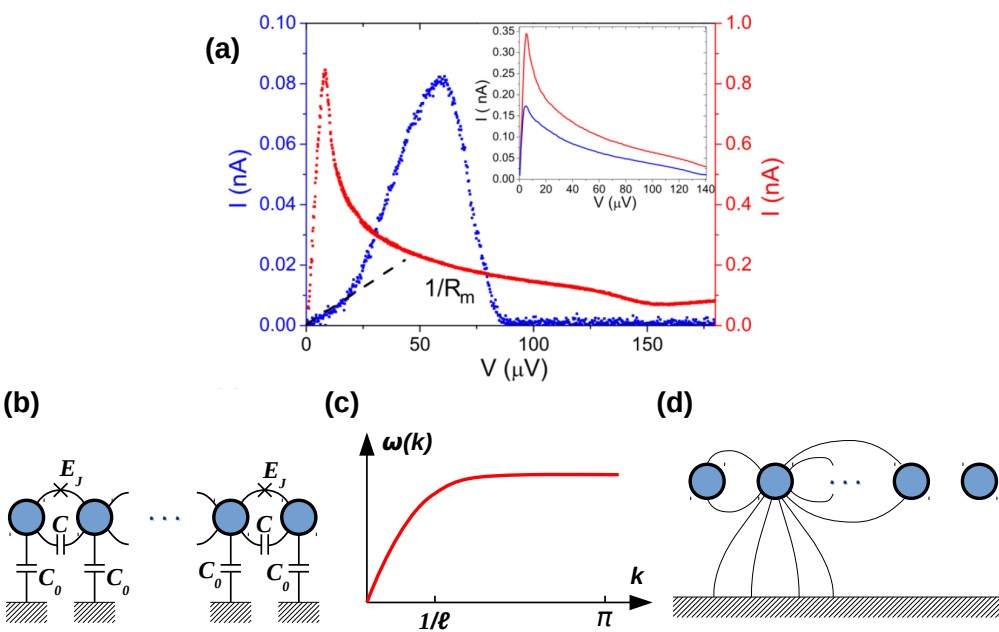

Figure 7: (a) Current-voltage characteristics of a small Josephson junction connected to a Josephson junction chain at two different values of the magnetic field. The inset shows the corresponding $I - V$ curve for the uniform chain [95]. (b) Schematic representation of a Josephson junction chain. Josephson and capacitive electrostatic coupling between neighboring superconducting islands are represented by the crosses and the capacitors $C$, respectively. The self-capacitance of the islands is represented by the capacitors $C_0$ between the islands and the ground. (c) Schematic plot of the frequency dispersion for phase excitations of a Josephson junction chain. (d) Schematic representation of the long-range Coulomb coupling between the chain islands and its screening by a remote ground plane (the thin curves represent electric field lines).

acquires an additional factor, $W \propto (N \text{ or } \sqrt{N})(N/\ell_s)^{-g}$. Here $g = \pi \sqrt{E_J/E_{C_0}}$ is determined by the ratio between the Josephson energy $E_J$ of each junction and the self-capacitance energy $E_{C_0} = (2e)^2/C_0$ of each island. Since typically $g \gtrsim 1$, the presence of the self-capacitance reverses the $N$ dependence of the QPS amplitude for long chains.

The factor $(N/\ell_s)^{-g}$, obtained in Ref. [97], strongly resembles the one obtained by Frank Hekking and Leonid Glazman much earlier [31] for the QPS amplitude on a Josephson junction included in a loop made of a superconducting wire (instead of $\ell_s$, another length scale appeared in that work). The reason for this similarity is that the island self-capacitance leads to screening of the long-range Coulomb interaction on the chain, thereby reinstating the Goldstone theorem and introducing low-energy phase modes in the system [Fig. (7(c)]. The properties of these modes are quite similar in superconducting wires and Josephson junction chains; it is the coupling of the phase slipping on one of the junctions to these delocalized low-energy phase modes that suppresses the QPS amplitude for long chains.

Having understood the importance of the low-energy modes in Josephson junction chains, Frank Hekking together with Denis Basko, Wiebke Guichard, Olivier Buisson and Nicolas Roch dedicated much effort to the theoretical description and experimental characterization of these modes. Since the Josephson coupling is nonlinear, the frequency of a given mode depends on

the number of quanta in all other modes. During Thomas Wiessl's PhD thesis, Kerr coefficients, which quantify this dependence, were calculated theoretically and measured experimentally, as was reported in Ref. [98]. A controllable way to introduce a non-linearity in a Josephson junction chain was proposed in Ref. [99]. Very recently, Frank Hekking and collaborators understood that in order to accurately model the mode dispersion, one may need to go beyond the simplistic representation of the Coulomb screening by ground capacitances, shown in Fig. 7 (b). Indeed, a nonlocal screening model, containing a new length scale, the distance between the chain and the nearby ground plane [Fig. 7 (d)], was shown to reproduce in a better way the experimentally measured mode frequencies [100].

In the last few years, Frank Hekking was actively promoting the idea of using spatially inhomogeneous Josephson junction chains as an environment whose properties could be engineered at will. Indeed, the fabrication technology allows to produce chains with each individual junction having specific parameters, which may be different from other junctions. Such spatial modulations would control the properties of the low-energy phase modes. The effect of random junction parameter variations was studied by Frank Hekking and Denis Basko in Ref. [101]. The simplest case of a spatial variation which can be purposefully introduced is a periodic modulation. Frank Hekking and coworkers calculated the Debye-Waller factor renormalizing the Josephson energy of a small junction included in the modulated chain [102] (the same effect for a spatially homogeneous superconducting wire was studied by Frank Hekking and Leonid Glazman in Ref. [31]). One of the last efforts led by Frank Hekking was the calculation of the QPS amplitude in a periodically modulated Josephson junction chain [103], building on the results for homogeneous systems [31, 97].

## 8   Mesoscopic physics with ultracold atoms

Frank Hekking was fascinated by the possibility of sharing the ideas and techniques of mesoscopic physics to understand cold atoms systems. Ultracold atoms are a very versatile system, where geometry and interaction strength can be tuned. For example, it is possible to realize a quasi-onedimensional geometry, ring traps and lattices. The magnitude and sign of interaction strength can be tuned. Furthermore, it is possible to follow the system during its real-time dynamics by non-destructive imaging methods. Frank Hekking had a clear vision that cold-atom quantum technology could be used to enlarge the typical domains of mesoscopic physics. He may be considered one of the founding leaders of Atomtronics [104], seeking to realize circuits of cold-atoms guided with laser light beams or magnetic means. Key aspects of the atomtronic systems are the charge-neutrality and the coherence properties of the fluid flowing in the circuits by applying synthetic gauge fields, the bosonic/fermionic statistics of the carriers, and the versatility of the operating conditions, thus allowing to conceive new quantum devices and simulators [105]. Important chapters in mesoscopic physics, like persistent currents in normal or superconducting rings, transport through quantum dots and more complex terminals, could be explored with a flexibility and control to a degree that was very hard to achieve with the traditional realizations of mesoscopic systems. The vision of Frank Hekking opened the way to an intense research activity, which is expected to have far-reaching implications both for mesoscopic and cold-atoms quantum technology.

In a first series of works in collaboration with Giulia Ferrini, Anna Minguzzi and Dominique Spehner, Frank Hekking has pushed forward the understanding of the quantum regime of a socalled Bose-Josephson junction. This is realized in its simplest form by a double well, hosting two Bose-Einstein condensates separated by a large barrier. Following an initial imbalance among the two wells, the system displays Josephson-like oscillations in the particle number imbalance among the two wells. If the interaction energy is comparable or larger than the

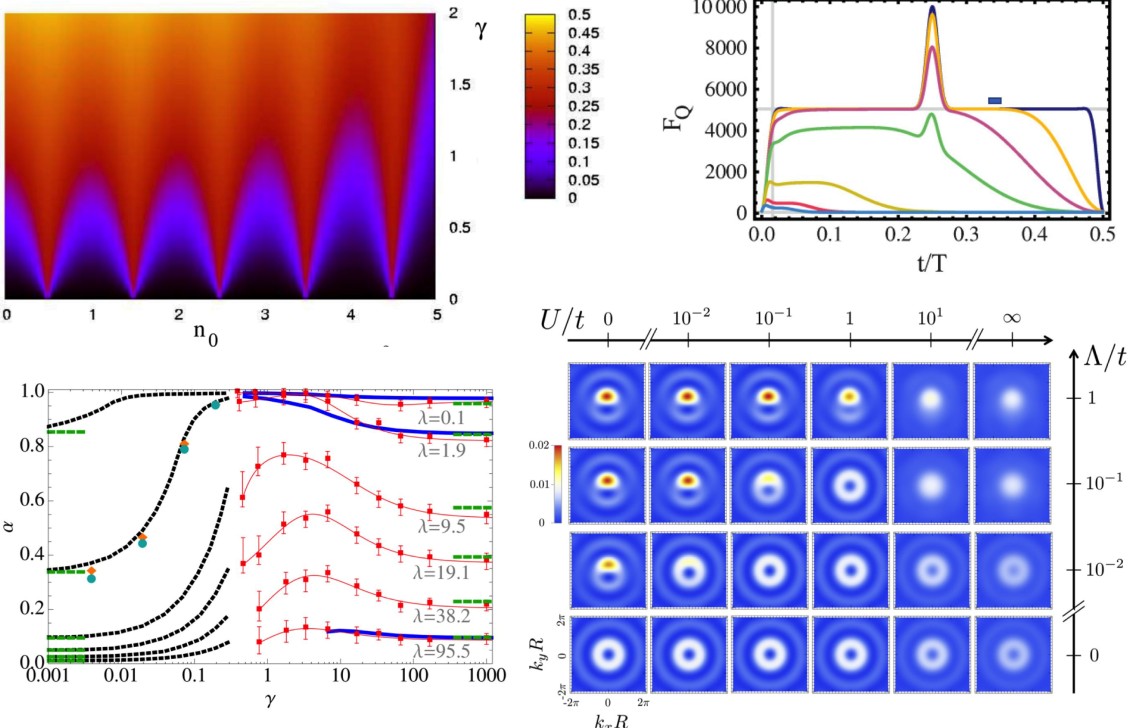

Figure 8: Top left: Number fluctuations in a Bose-Josephson junction as a function of the ratio $\gamma$ of tunnel to interaction energy (vertical axis), and imbalance among the two wells (horizontal axis) taken from [106]. Top right: Quantum Fisher information for a Bose-Josephson junction, as a function of time following a sudden quench of the tunnel energy to zero, at increasing phase noise [109]. Bottom left: Persistent current amplitude for one-dimensional bosons on a ring as a function of the interaction strength, for various values of barrier strength $\lambda$ [124]. Bottom right: Time-of-flight images of an expanding one-dimensional Bose gas on a lattice ring with barrier, carrying a superposition of angular momentum states, at varying interactions $U$ and barrier strength $\Lambda$ [128].

Josephson energy, the system enters a quantum regime where number squeezing occurs (see Fig. 8 top left). Furthermore, the dynamical evolution following a sudden quench of the Josephson energy to zero leads to the formation of macroscopic superposition states, which could be detected by using the concepts of full counting statistics [106, 107]. The creation of nonclassical states of the Bose-Josephson junction calls for the description of decoherence effects. Frank Hekking and coworkers have shown that some type of noise, such as fluctuations in the two wells, are relatively harmless for the Bose-Josephson junction, and their effect can be described exactly [108]. The two modes of the Bose-Josephson junction can be used for atom interferometry, where they can be mapped onto the two arms of the interferometer, where non-classical states could allow to outperform the classical limit. The effects of phase noise on quantum correlations useful for atom interferometry have been estimated in [109], by calculating the quantum Fisher information (see Fig. 8 top right).

Ultracold atoms in a tight waveguide behave as an effectively one-dimensional system once all the energy scales in the problem are smaller than the transverse confinement energy. In 2004, it has been demonstrated that bosons with repulsive interactions can be brought to the strongly-correlated regime [110, 111], where phase fluctuations dominate and the bosons display the phenomenon of statistical transmutation: at infinite interactions, in the so-called

Tonks-Girardeau regime [112], they share some properties as those of a Fermi gas. The strongly correlated regime can be theoretically described by a variety of techniques, ranging from the exact analytical solution at infinite interactions, to the Luttinger liquid theory at finite large interactions. In collaboration with Anna Minguzzi, Nicolas Didier and later with Guillaume Lang and Juan Polo, Frank Hekking has studied various mesoscopic aspects of the one-dimensional Bose gas. First, he focused on accurate expressions for first-order correlation functions for a finite-size system, showing that correlations decay with a universal power law with an exponent that depends on the proximity of the system's boundaries, namely the bulk and edge exponents are not the same [113]. Also, a careful analysis [114] of the large-distance decay of the correlations exhibit extra terms with respect to Haldane's original work [115], which are found in the exact solution at infinite interactions [116]. More recently, he also analyzed the properties of the density-density correlation function and the dynamic structure factor in connection with the concept of drag force in a one-dimensional fluid, studying in particular the effect of thermal fluctuations [117]. He also worked to an extension of the theory to a multi-mode quasi-1D geometry [118].

Another contribution of Frank Hekking and coworkers has been the study of the ground-state properties of the 1D Bose gas using the exact Bethe-Ansatz solution [119], where an extremely accurate analytical solution was proposed, capable of well describing the regime of intermediate interactions, where typically both weak-coupling and strong-coupling expansions fail. One of his very last works in this field has been the study of Josephson oscillations among tunnel coupled wires: in a 1D geometry, the Josephson oscillations of the population imbalance among the two wires are damped by an intrinsic bath of low-energy phonons, thus providing a realization of the Caldeira-Leggett model [120].

Progress in experiments with ultracold atoms allow nowadays to easily realize ring-shaped condensates. It is also possible to create a rotating, thin barrier in the ring. This naturally raises the question of the behaviour of the quantum fluid. In particular, ultracold 1D bosons share some superfluid properties with their higher-dimensional counterparts, which need a careful analysis in low dimensions and in finite systems as shown in a work in collaboration with Roberta Citro and Anna Minguzzi [121]. Ultracold atoms are essentially closed quantum systems, an ideal configuration to investigate quantum quenches, and non-equilibrium quantum dynamics. A sudden turn-on of the barrier velocity leads to oscillatory behaviour in the currents flowing in the system and the possibility of forming non-classical superpositions of angular momentum states, which can be studied exactly in the Tonks-Girardeau limit [122, 123].

More generally, the rotating barrier provides an example of an artificial gauge field which could be applied to the atoms, thus naturally leading to the question of the study of persistent currents for bosons. In a collaboration of the Grenoble group with Matteo Rizzi and Davide Rossini [124], it was shown that the amplitude of persistent currents depends on interaction and barrier strength in a non-monotonous way – the current amplitude is maximal at intermediate interactions (see Fig. 8 bottom left). This result was understood in terms of competition between the effects of classical screening at weak interactions and barrier renormalization by quantum fluctuations at large interactions. The concept of barrier renormalization actually applies to other physical observables and geometries: for example, it was shown that it affects the oscillation frequency of ultracold bosons in a harmonic trap split by a barrier [125], where an inhomogeneous Luttinger-liquid approach was developed and compared to an exact solution.

One of his last directions of investigation was the extension of the study of atomic rings beyond the strictly one-dimensional geometry. This was stimulated by exchanges with the experimental group of Hélène Perrin in Villetanneuse (Paris). The properties of a double ring lattice were studied in [126].

Together with Luigi Amico, Anna Minguzzi and coworkers, Frank Hekking studied a cor-

related Bose gas tightly confined into a ring shaped lattice, in the presence of an artificial gauge potential inducing a persistent current through it. A weak link created on the ring acts as a source of coherent back-scattering for the propagating gas, interfering with the forward scattered current. This system defines the atomic counterpart of the rf-SQUID: the Atomtronics Quantum Interference Device (AQUID). The full many-body Hamiltonian for interacting bosons on a lattice ring with gauge field was studied. For a special value of the gauge field, it was evidenced an anticrossing among the ground- and first-excited many-body energy levels, corresponding to a superposition of two angular momentum states in the system, giving rise to an effective two-level system. The dependence of the level splitting on lattice filling, interaction strength and weak-link strength was studied. As important output of this study, a clear avenue was provided in order to detect states with macroscopic phase coherence of clockwise and anti-clockwise cold-atoms currents [127, 128] (see Fig. 8 bottom right). AQUIDs have been defining a very interesting line of investigations both for experiment and theory [129–132]. Observing macroscopic quantum coherence and coherent quantum phase slips in experiments would provide major breakthroughs both in mesoscopic and cold-atoms physics.

## 9 Conclusion

In conclusion, we have reviewed a large part of Frank's Hekking scientific activity. Quite impressively, in an – alas too short – career, he has contributed to many domains of condensed matter and mesoscopic physics, as well as to the interfaces with mathematical and atomic physics. Low-dimensional conductors and superconductors, normal-superconductor junctions, heat transport and quantum thermodynamics, quantum phase slips, Josephson junctions are among the main keywords associated to his contributions. Unfortunately, the outline of this review did not allow us to describe a series of other works [133–140].

We hope that this review gives a faithful and somehow consistent picture of Frank Hekking's scientific work, and that it will remain as such in our memories.

## Acknowledgements

We thank the École de Physique des Houches for giving us the opportunity to organize the Frank Hekking Memorial Workshop in January 2018 as well as all the participants to this event. We also acknowledge the financial support of the Fondation Nanosciences Grenoble, the LabEx LANEF (ANR-10-LABX-51-01), the Center for Theoretical Physics in Grenoble, the University Grenoble Alpes. We sincerely thank Erika Borsje-Hekking for her help and support.

**Funding information** The recent activity of Frank Hekking was founded by Institut Universitaire de France, the ANR SuperRing (ANR-15-CE30-0012-02), the ANR QPS-NanoWires (ANR-15-CE30-0021-02).

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
