# Peer review of "Mesoscopic electron transport and atomic gases, a review of Frank W. J. Hekking's scientific work"

_SciPost Physics, doi:SciPost Phys. 5, 009 (2018)_

## Round 1 · Referee Report · Anonymous · 2018-5-5

Strengths

As the title indicates, this is a review of Frank Hekking's scientific work. It fulfils this purpose by providing concise and precise descriptions of his major contributions in 8 fields.

Weaknesses

There is a number of (mostly trivial) problems and typos etc. that I list in the "Report" and under "Requested changes" below.

Report

The only problems with the scientific content / readability are

p. 8 Fig. 3 Left: wrong arrow on the lowest vertical line

p. 10 "Cooper-pair sluice" is not common knowledge and needs a reference and/or an explanation

p. 23 "The full interacting many-body system was studied to evidence the two-level dynamics across different physical regimes of the system, as implied by the weak-link strength, interaction and filling fraction."
I don't understand this phrase, should be rewritten.

In my opinion, the paper is ready for publication once these problems have been addressed.

Requested changes

p. 2
effect of a weak disorder -> effect of weak disorder

Fig. 1middle -> Fig. 1 middle

p. 3
caption Figure 1: Left -> Left:

At finite temperature T -> At finite temperature $T$ (math mode)

p. 4
Atland -> Altland

through quantum wires [10], in a collaboration
-> through quantum wires [10] in a collaboration

p. 5
Blonder-Tinkham-Klapwijk -> Blonder-Tinkham-Klapwijk (BTK)

B. van Wees -> Bart van Wees

et al -> et al.

junction write [16,17]: -> junction can be written [16,17]

Eq. (1) diff -> \mathrm{diff}, remove dot after diff

the the tunnel -> the tunnel

R_diff: see above

low frequency current -> low-frequency current

voltages V -> voltages $V$ (math mode)

temperatures T -> temperatures $T$ (math mode)

Eq. (2): remove dot before I(V,T)

charge 2e, rather than e -> charge $2e$ rather than $e$

p. 7
Fluctuations corrections -> Fluctuation corrections

p. 8
caption Fig. 3
$\Delta$ is the superconducting ... -> $\delta$ is the level spacing
and $\Delta$ is the BCS superconducting gap

term non-local term -> non-local term

itself in the oscillations -> itself in oscillations

Fluctuations effects -> Fluctuation effects

current as a function -> the current as a function

On the opposite -> On the other hand

p. 9
to heat conductivity -> to the heat conductivity

low temperature limit -> low-temperature limit

of surface -> of a surface

p. 10
to Berry phase -> to the Berry phase

Majority of works -> The majority of works

from physics point of view -> from a fundamental point of view

or Cooper-pairs -> or Cooper pairs

and influence -> and the influence

p. 12
photon assisted tunneling -> photon-assisted tunneling

of high-temperature -> of a high-temperature

This device is formed -> This device consists of

in precise transfer -> in the precise transfer

with less influence -> with a reduced influence

is the one where -> is obtained if

to address energetics -> to address the energetics

p. 13
the superconducting qubits emergent field within
-> the emergent field of superconducting qubits in

Quantum-Electro Dynamics -> Quantum Electrodynamics

as Rydberg atom in high -> e.g., a Rydberg atom in a high

At our knowledge -> To our knowledge

this prediction was -> this was the first

to Cooper pair box -> to a Cooper-pair box

By current biasing -> By current-biasing

current biased -> current-biased

(ACPT) -> remove, is never used?

has been studied -> was studied

p. 14
an optimal control has been -> an optimal control scheme was

constituted by -> consisting of

has presented -> constitute

Didier PhD -> Didier's PhD

at al. -> et al.
(think about writing et al. in italics everywhere, \textit{et al.})

thermally activated behaviour -> thermally activated (TA) behaviour

phase diagram on -> phase diagram in

with various regimes -> in various regimes

showed that for a proper description of its dissipation
-> showed that dissipation

p. 15
caption Fig. 5:
oscillations frequency -> oscillation frequency

two level limit -> two-level limit

p. 16
junctions writes: -> junctions can be written as

power: -> power

scale plot of the I-V -> scale $I-V$ plot (math mode)

calculated together with Sukumar Rajauria
-> and Sukumar Rajauria calculated

p. 17
caption Fig. 6
where voltage -> where the voltage

against voltage -> against the voltage

Electron temperature -> The electron temperature

are supposed to -> are assumed to

phonons wavelength -> phonon wavelength

p. 18
hot-quasiparticles -> hot quasiparticles

in Ref. [85] was demonstrated -> in Ref. [85] it was demonstrated

as for example -> e.g.

as by using -> than by using

the 2000-2010 decade -> the decade 2000-2010

provided a theoretical support -> provided theoretical support

p. 19
caption Fig. 7
A schematic -> Schematic (3 times)

Self-capacitance -> The self-capacitance

in the current standard implementation
-> as implementations of a current standard

By that time -> At that time

p. 20
by ratio -> by the ratio

and self-capacitance -> and the self-capacitance

to theoretical -> to the theoretical

bulding up -> building on

p. 21
cold atoms quantum -> cold-atom quantum

strongly correlated -> strongly-correlated

p. 22
caption Fig. 8 ratio of tunnel to interaction
-> ratio $\gamma$ of tunnel energy to interaction energy

contain extra terms -> exhibit extra terms

to the original Haldane's work -> to Haldane's original work

of density-density -> of the density-density

in connection to -> in connection with

in the 1D geometry -> in a 1D geometry

p. 23
apply a rotating -> create a rotating

onto the ring -> in the ring

as their higher-dimensional -> with their higher-dimensional

Rossini, [123] -> Rossini [123]

for exemple -> for example
(or e.g.)

A weak link painted on -> A weak link created in

of such study -> of this study

for the experiments and theory -> for experiment and theory

in the experiments -> in experiments

p. 24 ff References
There are small problems in many references. I will list them in
increasing order of pedantry.

wrong page numbers etc.:
[7] 15 -> 3200

[9] Buttiker -> B\"uttiker

[12] 35 -> 1830

[16] 146 -> 6847

[17] title: 2 electrons -> two electrons

[20] 104 -> 4138

[28] volume number is 25, page number 721
last author D. Mailly is missing

[30] wrong doi, should be 10.1103/PhysRevB.46.9074

[51] M\"olmer -> M\o{}lmer

[76] D. Averin -> D.V. Averin

[114] is the only reference that contains two papers, is this intended?

[116] remove "6" after volume number 91

[119] abbreviate first names

[130] missing article number 4298

Finally,
- add missing doi in 11, 66, 103, 104, 109, 110, 111, 114, 115, 119,
125, 128, 129, 130, 131,

- remove "and" in all author lists to make the bibliography
consistent. Same for "et" in [53]. Put commas after each author name.

- add "0" as a first digit to the page numbers of

27, 36, 38, 42, 49, 61, 74, 77, 78, 90, 106, 107, 108, 112, 113, 117,
121, 123, 124, 127, 132, 133, 139

use consistent British or American spelling throughout the paper (?)

---

## Round 1 · Referee Report · Anonymous · 2018-5-25

Strengths

The paper is a complete and organic overview of the scientific actiity of Frank Hekking. Nearly all the works authored by Frank Hekking have been concisely, but accurately accounted for. The figures included improve the readability of the manuscript.

Weaknesses

I spot a few typos.

Report

The manuscript is sound and very well written. I recommend it for publication as is.

Requested changes

1) "asymptote" is misspelled in caption of fig. 1
2) caption of fig. 1, second to last line: $R_{Ss}$ to be replaced by $R_s$?
3) Line after Eq. (1): one needs to define the conductance $G_T$, which appear in Eq. (1), instead of (or in addition to) the resistance $R_T$
4) Third to last line in pag. 14: "... proper description of its dissipation and level quantization ..." replaced by "... proper description of it, dissipation and level quantization ..."
5) Typo "exemple" in pag. 23
6) Typo in the title of the paper [3], pag. 24

---

## Round 1 · Referee Report · Anonymous · 2018-5-25

Strengths

1- review the scientific legacy of one of the most eminent scientists in mesoscopic physics in the last decades
2- present an interesting historical account of the developments in the last decades
3- the summarised works offer an insight in how different branches of mesoscopic physics and cold atom physics have been developing in parallel and mutual benefit

Weaknesses

1- presents little new results, but this is expected for a legacy review
2- the references are a bit unbalanced and do not reflect the respective fields

Report

The article reviews the scientific work of Frank Hekking in the fields of mesoscopic electronics and cold atom physics. It provides not only an account of the work of an eminent scientists in the fields, but also demonstrates the interesting cross links between various topics established by Frank Hekking and his coworkers. Hence, I recommend it for publication.

Requested changes

1- a detailed list has been provided in another report and should be taken care of

---

## Round 2 · Author Response

Dear Editor,
Hereby we resubmit our manuscript for publication in Sci Post.
We sincerely thank the referees for their careful reading of our manuscript and their positive reviews.
We have made all the corrections suggested by the referees.
Thanking again for your support, best regards,
For the authors, Hervé Courtois

P.S.: I could not manage to remove the indication of a proceeding issue (Correlation function ...) in the above items. Please discard this.

---

## Round 2 · List of Changes

p. 2 effect of a weak disorder -> effect of weak disorder Fig. 1middle -> Fig. 1 middle

p. 3 caption Figure 1: Left -> Left: At finite temperature T -> At finite temperature T (math mode) caption of fig. 1, second to last line: RSs replaced by Rs "asymptote" corrected in caption of fig. 1

p. 4 Atland -> Altland through quantum wires [10], in a collaboration -> through quantum wires [10] in a collaboration

p. 5 Blonder-Tinkham-Klapwijk -> Blonder-Tinkham-Klapwijk (BTK) B. van Wees -> Bart van Wees et al -> et al. junction write [16,17]: -> junction can be written [16,17] Line after Eq. (1): define the conductance GT Eq. (1) diff -> \mathrm{diff}, remove dot after diff the the tunnel -> the tunnel R_diff: see above low frequency current -> low-frequency current voltages V -> voltages V (math mode) temperatures T -> temperatures T (math mode) Eq. (2): remove dot before I(V,T) charge 2e, rather than e -> charge 2e rather than e

p. 7 Fluctuations corrections -> Fluctuation corrections

p. 8 caption Fig. 3 Δ is the superconducting ... -> δ is the level spacing and Δ is the BCS superconducting gap term non-local term -> non-local term itself in the oscillations -> itself in oscillations Fluctuations effects -> Fluctuation effects current as a function -> the current as a function On the opposite -> On the other hand

p. 9 to heat conductivity -> to the heat conductivity low temperature limit -> low-temperature limit of surface -> of a surface

p. 10 to Berry phase -> to the Berry phase Majority of works -> The majority of works from physics point of view -> from a fundamental point of view or Cooper-pairs -> or Cooper pairs and influence -> and the influence

p. 12 photon assisted tunneling -> photon-assisted tunneling of high-temperature -> of a high-temperature This device is formed -> This device consists of in precise transfer -> in the precise transfer with less influence -> with a reduced influence is the one where -> is obtained if to address energetics -> to address the energetics

p. 13 the superconducting qubits emergent field within -> the emergent field of superconducting qubits in Quantum-Electro Dynamics -> Quantum Electrodynamics as Rydberg atom in high -> e.g., a Rydberg atom in a high At our knowledge -> To our knowledge this prediction was -> this was the first to Cooper pair box -> to a Cooper-pair box By current biasing -> By current-biasing current biased -> current-biased (ACPT) -> remove, is never used? has been studied -> was studied

p. 14 an optimal control has been -> an optimal control scheme was constituted by -> consisting of has presented -> constitute Didier PhD -> Didier's PhD at al. -> et al. (think about writing et al. in italics everywhere, \textit{et al.}) thermally activated behaviour -> thermally activated (TA) behaviour phase diagram on -> phase diagram in with various regimes -> in various regimes showed that for a proper description of its dissipation -> showed that dissipation Third to last line in pag. 14: "... proper description of its dissipation and level quantization ..." replaced by "... proper description of it, dissipation and level quantization ..."

p. 15 caption Fig. 5: oscillations frequency -> oscillation frequency two level limit -> two-level limit

p. 16 junctions writes: -> junctions can be written as power: -> power scale plot of the I-V -> scale I−V plot (math mode) calculated together with Sukumar Rajauria -> and Sukumar Rajauria calculated

p. 17 caption Fig. 6 where voltage -> where the voltage against voltage -> against the voltage Electron temperature -> The electron temperature are supposed to -> are assumed to phonons wavelength -> phonon wavelength

p. 18 hot-quasiparticles -> hot quasiparticles in Ref. [85] was demonstrated -> in Ref. [85] it was demonstrated as for example -> e.g. as by using -> than by using the 2000-2010 decade -> the decade 2000-2010 provided a theoretical support -> provided theoretical support

p. 19 caption Fig. 7 A schematic -> Schematic (3 times) Self-capacitance -> The self-capacitance in the current standard implementation -> as implementations of a current standard By that time -> At that time

p. 20 by ratio -> by the ratio and self-capacitance -> and the self-capacitance to theoretical -> to the theoretical bulding up -> building on

p. 21 cold atoms quantum -> cold-atom quantum strongly correlated -> strongly-correlated

p. 22 caption Fig. 8 ratio of tunnel to interaction -> ratio γ of tunnel energy to interaction energy contain extra terms -> exhibit extra terms to the original Haldane's work -> to Haldane's original work of density-density -> of the density-density in connection to -> in connection with in the 1D geometry -> in a 1D geometry

p. 23 apply a rotating -> create a rotating onto the ring -> in the ring as their higher-dimensional -> with their higher-dimensional Rossini, [123] -> Rossini [123] for exemple -> for example (or e.g.) A weak link painted on -> A weak link created in of such study -> of this study for the experiments and theory -> for experiment and theory in the experiments -> in experiments

p. 24 onwards, References Typo corrected in the title of the paper [3], pag. 24 [7] 15 -> 3200 [9] Buttiker -> B\"uttiker [12] 35 -> 1830 [16] 146 -> 6847 [17] title: 2 electrons -> two electrons [20] 104 -> 4138 [28] volume number is 25, page number 721, last author D. Mailly is added [30] doi corrected 10.1103/PhysRevB.46.9074 [51] M\"olmer -> M\o{}lmer [76] D. Averin -> D.V. Averin [114] splitted in two separate ones [116] remove "6" after volume number 91 [119] abbreviate first names [130] missing article number 4298 - add missing doi in 11, 66, 103, 104, 109, 110, 111, 114, 115, 119, 125, 128, 129, 130, 131, - remove "and" in all author lists to make the bibliography consistent. Same for "et" in [53]. Put commas after each author name. - add "0" as a first digit to the page numbers of 27, 36, 38, 42, 49, 61, 74, 77, 78, 90, 106, 107, 108, 112, 113, 117, 121, 123, 124, 127, 132, 133, 139

---

## Editorial Decision

published